# Perplexed by Perplexity: Perplexity-Based Data Pruning With Small Reference Models

**Zachary Ankner** [1,2]  **Cody Blakeney**[1]  **Kartik Sreenivasan**[1]
**Max Marion**[1]  **Matthew L. Leavitt**[3]  **Mansheej Paul**[1]
[1]Databricks  [2]MIT  [3]DatologyAI

## Abstract

In this work, we investigate whether small language models can determine high-quality subsets of large-scale text datasets that improve the performance of larger language models. While existing work has shown that pruning based on the perplexity of a larger model can yield high-quality data, we investigate whether smaller models can be used for perplexity-based pruning and how pruning is affected by the domain composition of the data being pruned. We demonstrate that for multiple dataset compositions, perplexity-based pruning of pretraining data can *significantly* improve downstream task performance: pruning based on perplexities computed with a 125 million parameter model improves the average performance on downstream tasks of a 3 billion parameter model by up to 2.04 and achieves up to a $1.45\times$ reduction in pretraining steps to reach commensurate baseline performance. Furthermore, we demonstrate that such perplexity-based data pruning also yields downstream performance gains in the over-trained and data-constrained regimes.

## 1 Introduction

A large focus of the machine learning community has been improving the performance of large language models (LLMs) while reducing their training costs. In this work, we consider how to improve the quality of an LLM by improving the quality of its pretraining data. Although there are many techniques to improve data quality, such as augmenting training samples with additional information (Li et al., 2024; Korbak et al., 2023), in this work we focus on the predominant method of *data pruning*: intelligently selecting a high-quality subset of a larger dataset to train on.

Data pruning is commonly used for quality filtering of noisy text data. Simple approaches include using symbolic rules (Bane et al., 2022; Raffel et al., 2020) or using simple classifiers to determine high-quality samples (Wenzek et al., 2020). However, in addition to basic quality filtering, more complex data pruning techniques are also applied to datasets to *further* improve their quality. Xie et al. (2023b) perform importance resampling where importance scores are calculated based on feature similarity to a target text. Tirumala et al. (2023) prune datasets by deduplicating and diversifying data based on a pretrained language model's embeddings of the text samples. Xie et al. (2023a) re-weight domain proportions based on learnability as determined by a smaller proxy model. Marion et al. (2023) investigate data pruning based on multiple neural heuristics of sample difficulty, ultimately concluding that the perplexity of a sample under a reference language model is the best pruning metric.

In this work, we thoroughly investigate the impact that data pruning based on sample perplexity (Marion et al., 2023) has on LLM pretraining. In particular, we focus on the interplay between pretraining dataset composition and pruning methodology. We further evaluate perplexity pruning in the over-trained and data-constrained regimes. We also investigate whether evaluating the quality of data interventions based on upstream test set perplexity is a sound methodology for gauging downstream performance. To perform perplexity-based data pruning, we train a small language model on a random subset of the given pretraining dataset and then evaluate its perplexity on each sample in the dataset. We then prune the dataset to only include samples within some range of perplexities (i.e., sub-sample to the highest or lowest perplexity samples). We demonstrate that for two vastly different pretraining data compositions, a small language model can be used to effectively prune the

pretraining dataset of a significantly larger model, leading to significant gains in the final model's downstream performance.

Our work differs from previous work on perplexity-based data pruning for LLM pretraining in three key ways: (i) our emphasis on downstream model quality evaluation, (ii) our exploration of different pretraining dataset domain compositions, and (iii) our analysis of pruning in non-standard training regimes. While previous works evaluate the resulting LLM's quality based on upstream metrics such as perplexity on the test split of the pretraining dataset, we evaluate data pruning's impact based on downstream evaluation benchmarks (e.g. *mmlu* (Hendrycks et al., 2021), *hellaswag*(Zellers et al., 2019), etc.). Evaluating on more meaningful benchmarks enables us to make stronger, more rigorous conclusions about the impact of perplexity-based data pruning, as we find that some techniques that significantly improve downstream performance have no, or even adverse, effects on upstream performance. This difference in metrics enables us to conclude that smaller models can prune the data for larger models, which was not observed in previous perplexity-based pruning works. Secondly, while previous work only investigates pruning on datasets composed of just one domain (CommonCrawl[1]), we consider two datasets with different domain compositions: the Pile (Gao et al., 2020) and Dolma (Soldaini et al., 2024). The Pile is composed of many diverse curated domains, with only 15.61% of the data being derived from general web-scrapes, while Dolma is a web-scrape skewed dataset, with 81.31% of its data being derived from the CommonCrawl. We find that successful pruning techniques vary greatly for different dataset compositions to the point that the best technique for one dataset composition may degrade performance for a different composition. Finally, we also evaluate perplexity-based data pruning in the less standard regimes of over-training and data-constrained training. This investigation provides a broader understanding for when practitioners should use perplexity pruning for their data.

**Contributions**    Our work makes the following contributions:

- We demonstrate that, across three datasets of varying domain compositions, a small reference model can effectively prune the pretraining dataset of a significantly larger language model ($30\times$ greater parameters), providing both a significant increase in downstream performance and decrease in pretraining steps (Table 1 and Figure 1).

- We show that data pruning techniques can be highly sensitive to the domain composition of the dataset, suggesting the need to evaluate multiple distinct dataset compositions when conducting data pruning research (Table 1 and Table 4).

- We investigate perplexity-based data pruning in multiple non-standard settings demonstrating that it can still lead to gains when over-training and when data-constrained (Section 3.4 and Section 3.5).

- We find that test set perplexity can be a misleading metric for evaluating the efficacy of data pruning techniques, as interventions that result in significantly higher test set perplexity can still achieve better performance on downstream tasks (Table 3).

## 2    PERPLEXITY-BASED DATA PRUNING

We start by training a reference model that will be used to calculate the perplexity of all samples in our dataset. First, we partition the original dataset into two splits: one for training the reference model and one for training the final model. After training the reference model on the standard next-token prediction objective, we compute the reference model's perplexity on each of the samples in the final model's training split. We then prune the final model's dataset split to a fraction of its original size, referred to as the *selection rate* ($r_s$), by selecting samples according to a *selection criteria* which can be one of low, medium, or high. In low selection, samples with the lowest perplexity are selected. In medium selection, we select samples whose perplexity is close to the median perplexity, that is, samples with perplexity in the $[50 - \frac{r_s}{2}, 50 + \frac{r_s}{2}]$ percentiles of all perplexities. In high selection, samples with the highest perplexity are selected. After pruning our dataset, we train a final model using the standard next token prediction objective on the pruned version of the final model training split. We present a pseudocode for pruning based on perplexity in Algorithm 1.

---

[1]https://data.commoncrawl.org/

---

**Algorithm 1:** Psuedocode for performing perplexity-based data pruning.

---

**Input:** Raw dataset $D = \{x^{(i)}\}_{i=1}^{M}$, where each $x^{(i)}$ is a tokenized text sample; `selection_criteria` $\in$ {low, medium, high}; selection rate $r_s \in (0,1)$; reference training split size $R$.

**Output:** Parameters of final model trained on the perplexity pruned dataset $\theta_{\text{final}}^*$.

$D_{\text{ref}}, D_{\text{train}} \leftarrow$ `random_split`$(D, R)$
$\theta_{\text{ref}} \leftarrow$ random parameter initialization
$\theta_{\text{ref}}^* \leftarrow$ `train`$(\theta_{\text{ref}}, D_{\text{ref}})$
`P` $\leftarrow \{\}$
**for** $x^{(i)} \in D_{train}$ **do**
    $\text{NLL}_{x^{(i)}} = \frac{1}{|x^{(i)}|} \sum_{t_j \in x^{(i)}} - \log P(t_j | t_{<j}; \theta_{\text{ref}})$
    $\text{PPLX}_{x^{(i)}} = 2^{\text{NLL}_{x^{(i)}}}$
    `P`$[x^{(i)}] = \text{PPLX}_{x^{(i)}}$
**end**
**if** `selection_criteria` $==$ *"low"* **then**
    `min_percentile` $\leftarrow 0.0$
    `max_percentile` $\leftarrow r_s$
**end**
**else if** `selection_criteria` $==$ *"medium"* **then**
    `min_percentile` $\leftarrow 0.5 - \frac{r_s}{2}$
    `max_percentile` $\leftarrow 0.5 + \frac{r_s}{2}$
**end**
**else if** `selection_criteria` $==$ *"high"* **then**
    `min_percentile` $\leftarrow 1 - r_s$
    `max_percentile` $\leftarrow 1.0$
**end**
$\hat{F}_P \leftarrow$ empirical CDF of `P.values()`
$D_{\text{pruned}} \leftarrow []$
**for** $x^{(i)}, PPLX_{x^{(i)}} \in P$ **do**
    **if** `min_percentile` $< \hat{F}_P(PPLX_{x^{(i)}}) <$ `max_percentile` **then**
        $D_{\text{pruned}}$`.append`$(x^{(i)})$
    **end**
**end**
$\theta_{\text{final}} \leftarrow$ random parameter initialization
$\theta_{\text{final}}^* \leftarrow$ `train`$(\theta_{\text{final}}, D_{\text{pruned}})$
**return** $\theta_{final}^*$

---

We consider the setting in which the reference model is significantly smaller than the final model. While this assumption is not strictly necessary, we believe that it is the most practically relevant setup, as it best reflects a data pruning paradigm that would be used for the next generation of LLMs where the models being trained are larger than any existing models.

## 3 EXPERIMENTS

### 3.1 SETUP

**Models.** All models are based on the MPT family of transformer models (Vaswani et al., 2017; MosaicML, 2023c). All reference models have 125 million parameters, and we consider final models with 1 billion and 3 billion parameters.

**Data.** We consider two datasets in this work. The Pile (Gao et al., 2020) is composed of 22 different domains that range from general web scrapes to legal text. Dolma (Soldaini et al., 2024) is composed of 7 different domains and is derived mainly from general web scrapes. We tokenize all datasets using the GPT-4 tokenizer (OpenAI, 2022).

Table 1: Average normalized accuracy grouped by task category for both datasets and both final model sizes. For all datasets and model sizes we find that training on perplexity pruned data outperforms the baseline. Bold results are within one standard error of the highest score.

| Pruning Method | World Knowl-edge | Common Sense Reason-ing | Language Under-stand-ing | Symbolic Prob-lem Solving | Reading Com-prehen-sion | Average |
|---|---|---|---|---|---|---|
| **1B Parameters Trained on Pile** | | | | | | |
| No Pruning (Baseline) | 15.51 | 10.31 | 28.11 | **3.53** | **11.16** | 13.73 |
| High Perplexity Selected | **18.18** | **12.75** | **33.2** | 3.36 | **10.63** | **15.62** |
| **3B Parameters Trained on Pile** | | | | | | |
| No Pruning (Baseline) | 21.82 | 13.09 | 39.08 | **4.88** | 14.28 | 18.63 |
| High Perplexity Selected | **25.8** | **16.24** | **43.32** | 2.91 | **15.07** | **20.67** |
| **1B Parameters Trained on Dolma** | | | | | | |
| No Pruning (Baseline) | 16.48 | 12.32 | 28.86 | **3.58** | 7.95 | 13.84 |
| Medium Perplexity Selected | **17.98** | **13.03** | **31.87** | **3.44** | **10.41** | **15.35** |
| **3B Parameters Trained on Dolma** | | | | | | |
| No Pruning (Baseline) | 23.56 | 14.29 | 39.57 | **4.4** | **14.2** | 19.2 |
| Medium Perplexity Selected | **24.19** | **16.48** | **41.8** | 3.3 | 13.19 | **19.79** |

**Training and hyperparameters.** All reference models are trained for a fixed duration of 26 billion tokens. Unless otherwise specified, all final models are trained to Chinchilla optimal (Hoffmann et al., 2022), meaning that each final model's training duration in tokens is 20 times its parameter count. All models are trained using the decoupled Lion optimizer (Chen et al., 2024) with a cosine learning rate schedule. All reference models and 1B parameter models are trained with a maximum learning rate and weight decay of `2e-4` and all 3B models are trained with a maximum learning rate and weight decay of `1.6e-4`. Training is conducted using llm-foundry (MosaicML, 2023b) and using both Nvidia `A100s` and `H100s`. We perform two trials for each experiment.

**Evaluation.** We evaluate models on 33 different downstream question-answering tasks using the MosaicML evaluation gauntlet (MosaicML, 2023a). Before averaging the accuracy across tasks, we normalize the accuracy on each task by the baseline of random guessing Specifically, we normalize the accuracy of each individual task as $a_n = \frac{a_m - a_r}{1 - a_r}$, where $a_m$ is the accuracy of the model and $a_r$ is the expected accuracy of random guessing. We report the average normalized accuracy for each task category as well as the average normalized accuracy across all task categories. [2]. More details on tasks and task categories are listed in Section 9.

## 3.2 PERPLEXITY-BASED DATA PRUNING IMPROVES DOWNSTREAM PERFORMANCE

If a certain range of perplexities is a good heuristic for data quality, training on that perplexity-pruned subset should improve downstream performance. We sweep across pruning selection criteria and selection rates (Section 7) and find that the best settings are to select high-perplexity samples at a 50% rate for the Pile and to select medium-perplexity samples at a 50% rate for Dolma. We compare the most performant pruning settings to baseline models trained on the original datasets without pruning in Table 1. Across all datasets and model sizes, models pretrained on the perplexity pruned version of the dataset significantly outperform the baseline model on average. Specifically, perplexity-based data pruning outperforms the average downstream performance of no pruning for 1B models by 1.89 and 1.51 for the Pile and Dolma respectively, and improves the performance of 3B models by 2.04 and 0.59 for the Pile and Dolma respectively. These results suggest that the perplexity of a small model provides a strong signal of data quality for a much larger model, as training on the data selected by the small model leads to significant downstream performance improvements.

---

[2]Not to be confused, the random accuracy normalization we use is different from the normalized accuracy reported by the EleutherAI LM Evaluation Harness, which normalizes based on the Byte-length of the response.

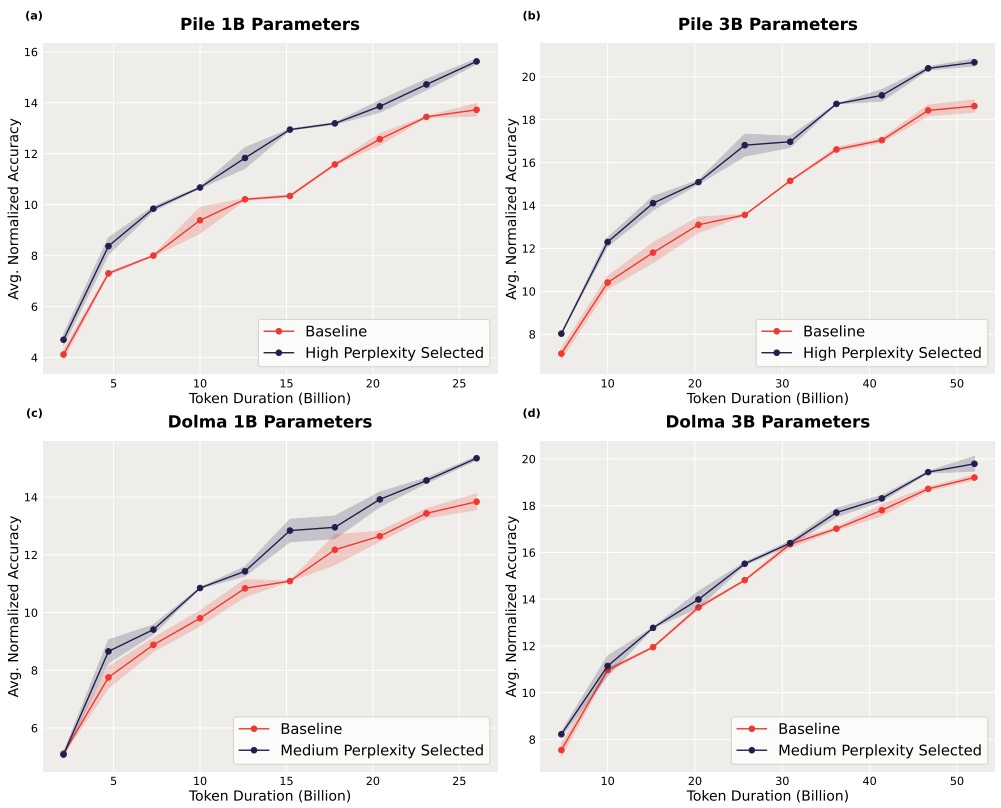

Figure 1: Average normalized task accuracy evaluated intermittently throughout pretraining for each dataset and model size investigated. Perplexity-based data pruning leads to an improvement in performance for all intermediate training steps evaluated.

### 3.3 PERPLEXITY-BASED DATA PRUNING IMPROVES TRAINING EFFICIENCY

Since perplexity-based data pruning improves the final performance of models, we also investigate how pruned data affects the training dynamics of models. Specifically, we investigate whether training on perplexity pruned data enables models to achieve the same downstream performance as models trained on the unpruned data in training fewer steps. We plot the average downstream performance of partially trained checkpoints from the 1B baseline and perplexity pruned models in Figure 1. Perplexity pruning outperforms the baseline model for all intermediate pretraining durations evaluated. Furthermore, perplexity pruned models reach the same average normalized accuracy as the baseline models in $1.31\times$ and $1.45\times$ fewer steps for Pile 1B and 3B respectively and in $1.29\times$ and $1.14\times$ fewer steps for Dolma 1B and Dolma 3B respectively. These results demonstrate that the resulting high-quality data from perplexity-based data pruning enables faster learning which can be leveraged to achieve the same downstream performance as training on unpruned data with fewer pretraining steps.

### 3.4 PERPLEXITY-BASED DATA PRUNING FOR OVER-TRAINED MODELS

A recent trend with LLMs has been to over-train models by training them on more tokens than the Chinchilla optimal number of tokens (Touvron et al., 2023; Gadre et al., 2024). As our work targets the data component of LLM pretraining, we investigate the hypothesis that over-training would be more beneficial for models trained on perplexity pruned datasets as the data is of higher quality. We test this hypothesis by training a 1B parameter model for 130B tokens, which is $5\times$ the Chinchilla optimal number of tokens. We evaluate the downstream performance of each over-trained model in Table 2. The main observation is that while the absolute gain in average downstream normalized accuracy from perplexity-based data pruning on the Pile is similar for both compute optimal and

Table 2: Downstream task performance for Chinchilla Optimal and $5\times$ over-trained data budgets. The "Improvement Over Baseline" column refers to the gain observed from perplexity pruning as compared to the baseline trained in the same setting.

| Pruning Method | Average | Improvement Over Baseline |
|---|---|---|
| **1B Parameters Trained on High Perplexity Pile** | | |
| Chinchilla Optimal | 15.62 | 1.89 |
| $5\times$ Over-Trained | 18.83 | 1.74 |
| **1B Parameters Trained on Medium Perplexity Dolma** | | |
| Chinchilla Optimal | 15.35 | 1.51 |
| $5\times$ Over-Trained | 18.67 | 0.84 |

over-trained models, the gain decreases for Dolma when over-training. On the Pile, we find that the gain from perplexity pruned data is similar in the compute optimal regime and the over-trained regime: we see a gain in average performance of 1.89 when training compute optimal and a gain of 1.74 when over-training. On Dolma, the gain from perplexity pruned data decreases in the over-trained regime: we see a gain of 1.51 when training for a compute optimal duration but this decreases to a gain of 0.84 when over-training. These results show that while the higher quality data resulting from perplexity-based data pruning does still lead to an improvement in downstream performance in the over-trained regime, there is not a relative increase in downstream improvement over the baseline when over-training.

## 3.5 PERPLEXITY-BASED DATA PRUNING FOR THE DATA CONSTRAINED REGIME

Our experiments so far were conducted in the setting where there exists a sufficient abundance of data such that even after pruning with the desired selection rate there are enough data points to fill the desired token budget without requiring any data to be repeated. However, there are many training settings that do not fall under this data-abundant regime. Consequently, we evaluate how perplexity-based data pruning performs when the number of tokens is constrained, and pruning induces a greater number of repetitions of the data. For each dataset, we vary the available data such that training for a Chinchilla optimal number of tokens requires a different number of repetitions. Specifically, we investigate data budgets that require $\{0.5, 1, 2, 4, 8\}$ repetitions to reach the Chinchilla optimal number of tokens[3]. As each number of repeats refers to the total number of tokens available, for all pruning experiments the number of repetitions after pruning is actually greater by a factor of $\frac{1}{r_s}$ since we prune the available tokens according to $r_s$, the selection rate. Since all models use a selection rate of 0.5, the models trained on the pruned data see the data for $2\times$ more repetitions.

We plot the average downstream performance as a function of the number of repetitions in Figure 2. On both the Pile and Dolma, we find that training on perplexity pruned data yields an improvement for up to two repetitions. These results suggest that perplexity-based data pruning can still provide performance gains for some degree of data constraint. Furthermore, our results replicate the findings of Muennighoff et al. (2023) that more than four repetitions yields negligible gains. Specifically, the baseline model without pruning maintains commensurate performance for up to four repetitions. Similarly, models trained on perplexity-pruned data maintain commensurate performance for up to two repetitions through the base data, which corresponds to four repetitions after pruning. That training on repeated perplexity-pruned data leads to diminishing gains after four repetitions post-pruning suggests that the higher quality data resulting from pruning does not change the point for which repeating data yields diminishing improvements in performance.

## 3.6 UPSTREAM PERPLEXITY IS NOT A RELIABLE EVALUATION METRIC FOR DATA PRUNING

As previous works have used the perplexity of the model on a test split of the pretraining dataset as an approximation to downstream performance, we wanted to explore how well such perplexity-based

---

[3]Repeat=0.5 means that the available number of tokens is twice the training budget, i.e. the data-abundant setting

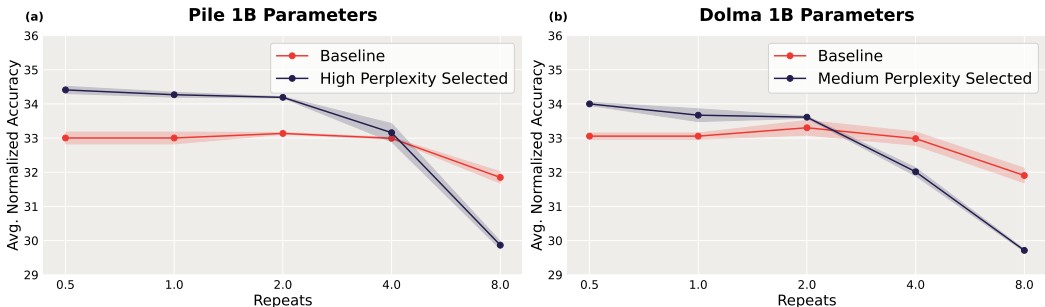

Figure 2: Downstream task performance as a function of available dataset size. The number of repeats denotes the number of repeats over the raw dataset necessary to achieve the Chinchilla optimal number of tokens. Training on perplexity pruned data leads to an improvement for up to two repeats on both the Pile Dolma.

Table 3: Performance as evaluated by perplexity on a test split of the original dataset as well as average normalized task accuracy for 1 billion parameter final models trained on the Pile. The model trained on pruned data has worse pretraining test split perplexity even though it significantly improves average downstream normalized accuracy.

| Pruning Method | Test Set Pplx. ($\downarrow$) | Downstream Task Avg. ($\uparrow$) |
|---|---|---|
| 1B Parameters Trained on Pile | | |
|   No Pruning (Baseline) | **7.83** | 13.73 |
|   High Perplexity Selected | 8.51 | **15.62** |
| 1B Parameters Trained on Dolma | | |
|   No Pruning (Baseline) | **13.53** | 13.84 |
|   Medium Perplexity Selected | 14.33 | **15.35** |

evaluations agree with downstream performance for data intervention techniques. Pruning performs an intervention on the dataset, making models trained on the pruned dataset biased estimators of the original data distribution. Therefore, it is unlikely that the performance on the original data distribution is a fair evaluation of model quality. We compare the test set perplexity and average downstream performance for 1 billion parameter models trained on the original and pruned version of the Pile and Dolma in Table 3. For both the Pile and Dolma, training on perplexity pruned data significantly worsens perplexity on a test split of the pretraining data, while the average downstream performance is significantly improved. This result suggests that test set perplexity may not always be a sound metric for data pruning work and that researchers should instead directly evaluate on downstream benchmarks.

# 4 Understanding the Effects of Perplexity-Based Pruning

In this section, we investigate how data pruning works by exploring some of the properties of perplexity-based pruning.

## 4.1 How Are Reference Perplexities Distributed

In order to better understand how perplexity-based data pruning works, we investigate the distribution of the computed reference model perplexities for each dataset. For each dataset, we randomly sample 10% of the calculated perplexities and perform kernel density estimation to estimate the distribution of log perplexities for a given dataset. We repeat this procedure for the optimal pruned version of the dataset. We plot the resulting estimates of the log perplexity distribution in Figure 3. We find that the log perplexity distribution for the Pile is multimodal and asymmetric, while for Dolma and it is unimodal and symmetric.

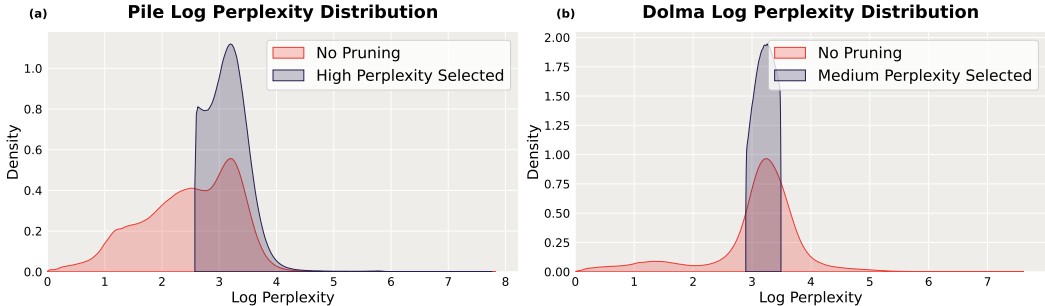

Figure 3: Distribution of sample perplexities as evaluated by the reference model for the Pile and Dolma. We show both the original distribution over the full dataset without pruning as well as the distribution after applying the optimal perplexity-based data pruning technique for a given dataset.

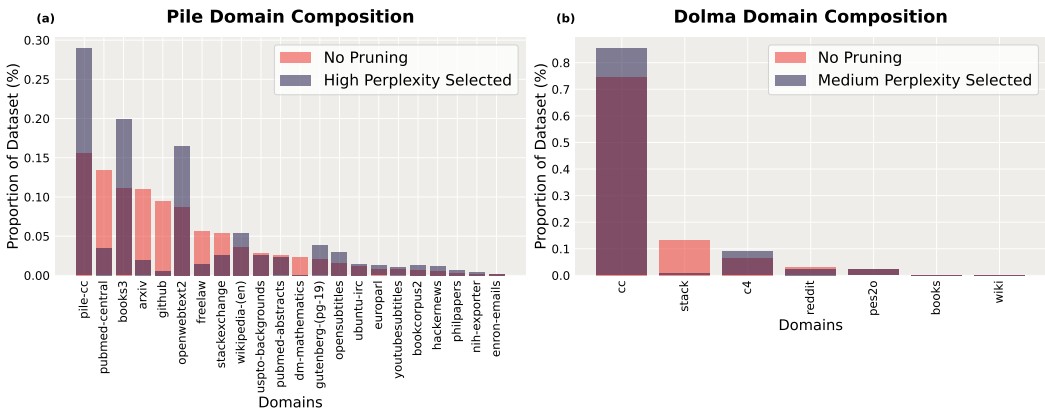

Figure 4: Proportion of the total dataset each domain makes up before and after pruning. For all datasets, pruning tends to select more samples from general web domains while leaving out samples from highly specific domains.

## 4.2   HOW PRUNING AFFECTS DOMAIN COMPOSITION

We can also interpret the effect that perplexity-based data pruning has on a dataset by examining how pruning affects each domain's proportion of the total dataset. We plot the pre and post-pruning domain compositions for the Pile and Dolma in Figure 4. Interestingly, for all datasets pruning increases the proportion of data coming from web-scraped domains while decreasing the proportion of data coming from highly specific technical domains such as code or scientific papers. This trend is more pronounced in the Pile, where the proportions of Pile-CC and OpenWebText2 nearly double, while the proportions of domains such as Pubmed Central, ArXiv, and Github are all reduced by at least a factor of three. Future work should investigate how perplexity-based pruning affects a model's performance on downstream tasks that are in the same category as the highly pruned domains.

## 5   RELATED WORK

**Classical methods for pruning text data.**   In order to improve the quality of raw web scrapes, which often contain very noisy samples, pruning via quality filtering has become a common practice. Simple rules-based methods have been employed to prune datasets by filtering out low-quality samples according to some hand-crafted heuristic such as whether the text contains prohibited words, is predominantly English, etc. (Bane et al., 2022; Raffel et al., 2020; Rae et al., 2022; Penedo et al., 2023). N-gram perplexity-based methods, in which an n-gram model is first trained on a high quality, curated corpus and then used to score another corpus, have also been applied to filter text

data (Moore & Lewis, 2010; Axelrod, 2017; Gao, 2021; Laurençon et al., 2022; Muennighoff et al., 2023). Although our method also uses perplexity to prune data, it does so in a very different manner. In n-gram perplexity pruning, perplexity is used to estimate whether new text is in distribution as compared to the currated text the n-gram was trained on, while in our model-based perplexity pruning, the reference model is trained on the same distribution of text and the perplexity is more akin to an estimate of the difficulty of an example. In this work, the datasets we leverage already have some basic rules-based pruning applied, and as such, the method we investigate is largely complementary to these existing techniques.

**Neural network based methods for pruning text data.**   Recently, there has been much interest in using neural networks to compute metrics that can be used to intelligently prune datasets. A common technique in this family of methods is using a model to sample high-quality data from large datasets based on the sample's similarity to a curated high-quality corpus that serves as a target distribution (Feng et al., 2022; Xie et al., 2023b). Xie et al. (2023a) also consider how to use a small reference model to prune pretraining data for a much larger model, by using a small reference model to learn the optimal weighting of domain proportions to maximize the "learnability" of the resulting dataset. Pruning based on the difficulty or loss of a sample has previously been explored for text data, but the majority of such work focuses on curating data for finetuning (Swayamdipta et al., 2020; Attendu & Corbeil, 2023; Coleman et al., 2020; Mindermann et al., 2022; Mekala et al., 2024). Marion et al. (2023); Yang et al. (2025); Sachdeva et al. (2024), however, investigate multiple model-based sample difficulty heuristics for pruning pretraining text datasets. Although we use the same method for pruning text pretraining datasets as Marion et al. (2023), our analysis differs substantially as we evaluate model quality based on downstream metrics and extend our analysis to multiple different dataset compositions which enables us to conclude that the reference model can be smaller than the final model.

**Data pruning on vision tasks.**   While data pruning is becoming more and more relevant with large amounts of text data, it has also been extensively applied in the vision domain (Paul et al., 2021; Toneva et al., 2018; Park et al., 2023). These works often prune data points based on their loss or gradients during training (Killamsetty et al., 2021; Mirzasoleiman et al., 2020). Model-based methods have also been leveraged for image data pruning (Fang et al., 2024; Schuhmann et al., 2021). Note that in the literature, data pruning is also sometimes referred to as coreset selection (Guo et al., 2022). More recently, Park et al. (2022) show that, somewhat surprisingly, active learning (Castro & Nowak, 2008) based algorithms tend to outperform most data subset selection algorithms. In the context of contrastive learning, hard-negative mining has been effective as a data pruning method (Kalantidis et al., 2020; Robinson et al., 2020; Zhang & Stratos, 2021). Recently, Goyal et al. (2024) investigated scaling laws for training on pruned data in the context of vision models.

## 6    CONCLUSION

In this work, we conduct an empirical investigation of the impact that perplexity-based data pruning has on model performance. We demonstrate that small reference models can be used to prune the data of models with up to $30\times$ more parameters, leading to both significant downstream performance improvements and increased training efficiency. We then investigate perplexity-based data pruning in two non-standard settings: the over-trained and data-constrained regimes. We find that for both settings, training on perplexity pruned data can outperform training on unpruned data, demonstrating that perplexity-based data pruning is a widely applicable and extensible technique. We also investigate upstream metrics for evaluating data pruning techniques and provide an example where evaluating models based on their perplexity on the test split of the pretraining dataset does not align with evaluating based on downstream model performance. One limitation of our work is that while domains such as code are largely pruned, we do not have the computational budget required for training models of sufficient size to test the coding ability of models on benchmarks such as HumanEval (Chen et al., 2021). It remains important future work to scale perplexity pruning to larger models such that the impact of pruning specific domains can be better understood. Our work takes a key step towards establishing perplexity-based data pruning as a technique in the modern data researcher's toolkit.

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

Table 4: Results from sweeping different selection criteria. We report the average normalized accuracy for each task grouping as well as across all tasks. While high perplexity selection is optimal for the Pile, medium perplexity selection is optimal for Dolma. Bold results are within one standard error of the highest normalized accuracy.

| Pruning Method | World Knowl­edge | Common Sense Reason­ing | Language Under­stand­ing | Symbolic Prob­lem Solving | Reading Com­prehen­sion | Average |
|---|---|---|---|---|---|---|
| 1B Parameters Trained on Pile | | | | | | |
| No Pruning (Baseline) | 15.51 | 10.31 | 28.11 | **3.53** | **11.16** | 13.73 |
| Low Perplexity Selected | 11.14 | 5.76 | 18.66 | **3.54** | 8.72 | 9.56 |
| Medium Perplexity Selected | 16.12 | 9.01 | 28.1 | **3.41** | **10.86** | 13.5 |
| High Perplexity Selected | **18.18** | **12.75** | **33.2** | **3.36** | **10.63** | **15.62** |
| 1B Parameters Trained on Dolma | | | | | | |
| No Pruning (Baseline) | 16.48 | 12.32 | 28.86 | **3.58** | 7.95 | 13.84 |
| Low Perplexity Selected | 16.13 | 10.1 | 27.28 | **3.45** | 7.85 | 12.96 |
| Medium Perplexity Selected | **17.98** | **13.03** | **31.87** | **3.44** | **10.41** | **15.35** |
| High Perplexity Selected | 16.65 | **13.12** | **31.14** | 3.15 | 8.55 | 14.52 |

Yu Yang, Siddhartha Mishra, Jeffrey Chiang, and Baharan Mirzasoleiman. Smalltolarge (s2l): Scalable data selection for fine-tuning large language models by summarizing training trajectories of small models. *Advances in Neural Information Processing Systems*, 37:83465–83496, 2025.

Rowan Zellers, Ari Holtzman, Yonatan Bisk, Ali Farhadi, and Yejin Choi. Hellaswag: Can a machine really finish your sentence? In *Proceedings of the 57th Annual Meeting of the Association for Computational Linguistics*, 2019.

Wenzheng Zhang and Karl Stratos. Understanding hard negatives in noise contrastive estimation. *arXiv preprint arXiv:2104.06245*, 2021.

# 7 FULL DATA PRUNING SETTINGS SWEEP

In this section, we report the results of sweeping over different perplexity-based pruning setting configurations. In particular, for each dataset, we first sweep over the selection criteria to determine where from the distribution of perplexities samples should be selected. Then, using the best selection criteria, we sweep the selection rate to determine how much we should prune.

**Setup.** We use the same training and evaluation setup as detailed in Section 3.1. We only perform the sweep over pruning settings for 1 billion parameter final models for computational budget reasons; however, we find that the best selection criteria at the 1 billion parameter scale also confers a performance improvement at the 3 billion parameter scale, as detailed in 3.2.

## 7.1 FINDING THE BEST SELECTION CRITERIA

For each dataset, we first sweep the selection criteria while keeping the selection rate fixed at 50%. We report the performance of each selection criteria in Table 4. We find that on the Pile high perplexity selection works the best and on Dolma medium perplexity selection works the best, improving the average downstream performance by 1.89 and 1.51 respectively. An important observation from the sweep is that the best selection criteria from one dataset does not transfer to another dataset and may actually degrade performance compared to the baseline. Although medium-perplexity selection is the best method on Dolma, selecting medium-perplexity samples on the Pile leads to a decrease in the average downstream performance of 0.23 as compared to not performing pruning. These results inform us that high and medium perplexity selection are the optimal selection criteria for the Pile and Dolma respectively, and that the optimal selection criteria does not necessarily transfer between datasets with different domain compositions.

Table 5: Results from sweeping different selection rates. We report the average normalized accuracy for each task grouping as well as across all tasks. Bold results are within one standard error of the highest normalized accuracy.

| Pruning Method | World Knowl- edge | Common Sense Reason- ing | Language Under- stand- ing | Symbolic Prob- lem Solving | Reading Com- prehen- sion | Average |
|---|---|---|---|---|---|---|
| 1B Parameters Trained on Pile | | | | | | |
| 25% Selection Rate | **18.21** | **12.88** | **34.44** | **3.73** | **9.44** | **15.74** |
| 50% Selection Rate | **18.18** | **12.75** | 33.2 | 3.36 | **10.63** | **15.62** |
| 75% Selection Rate | 17.08 | 10.11 | 31.37 | **3.81** | 9.02 | 14.28 |
| 1B Parameters Trained on Dolma | | | | | | |
| 25% Selection Rate | 17.94 | **12.16** | **31.63** | 3.58 | 8.91 | 14.85 |
| 50% Selection Rate | **17.98** | **13.03** | **31.87** | 3.44 | 10.41 | **15.35** |
| 75% Selection Rate | **18.2** | 11.78 | 29.96 | **3.32** | **10.82** | 14.82 |

## 7.2 FINDING THE BEST SELECTION RATE

Using the optimal selection criteria that we found for each dataset, we next investigate the best selection rate for each dataset. We investigate three different selection rates: 25%, 50%, and 75%. We present the results for each selection rate in Table 5. On the Pile, we find that there is no significant difference in downstream performance for selection rates of 25% and 50%; on Dolma we find that a selection rate of 50% achieves the best average downstream performance. For simplicity, we chose to conduct the rest of the experiments in the paper using a selection rate of 50% on both datasets. Furthermore, we find that all the selection rates tested outperform the baseline of no data pruning as measured by average downstream performance. This suggests that the selection criteria has a greater impact on the performance of a pruning configuration than the selection rate.

## 8 EFFICIENCY ANALYSIS

We now turn towards the total computational complexity of perplexity pruning. As our method does not change the computation performed on the forward or backward pass, and only affects which tokens are trained on, we can analyze the compute requirements in terms of the total number of operations. We will approximate the cost of training as $6ND$ where $N$ is the number of parameters and $D$ is the number of training tokens, and we will approximate the cost of computing a sequence's perplexity in inference mode as $2ND$ (Kaplan et al., 2020; Anthony et al.). Assuming a reference model of size $N_{\text{ref}}$, number of reference training tokens $D_{\text{ref}}$, final model size $N_{\text{final}}$, number of final tokens $D_{\text{final}}$, selection rate $r_s$, and fraction of tokens for the pruned data to achieve the same performance as the baseline $F$, the relative cost of perplexity pruning compared to the baseline is:

$$\frac{2N_{\text{ref}}\frac{D_{\text{final}}}{r_s} + 6N_{\text{ref}}D_{\text{ref}} + 6N_{\text{final}}FD_{\text{final}}}{6N_{\text{final}}D_{\text{final}}} = \frac{N_{\text{ref}}}{N_{\text{final}}}\left(\frac{1}{3r_s} + \frac{D_{\text{ref}}}{D_{\text{final}}}\right) + F$$

All our reference models are 125M parameters, trained for 26B tokens, and we use $r_s = 0.5$ throughout. The 1B models are trained on 26B tokens giving us $\frac{N_{\text{ref}}}{N_{\text{final}}}\left(\frac{1}{3r_s} + \frac{D_{\text{ref}}}{D_{\text{final}}}\right) = \frac{125\times10^6}{1.3\times10^9}\left(\frac{2}{3} + \frac{26\times10^9}{26\times10^9}\right) = 0.16$. For the Pile, $F = 0.76$ giving us a relative cost of $0.16 + 0.76 = 0.92$ and for Dolma $F = 0.78$ giving us a relative cost of $0.16 + 0.78 = 0.94$.

The 3B models are trained on 54B tokens giving us $\frac{N_{\text{ref}}}{N_{\text{final}}}\left(\frac{1}{3r_s} + \frac{D_{\text{ref}}}{D_{\text{final}}}\right) = \frac{125\times10^6}{2.7\times10^9}\left(\frac{2}{3} + \frac{26\times10^9}{54\times10^9}\right) = 0.05$. For the Pile, $F = 0.69$ giving us a relative cost of $0.69 + 0.05 = 0.74$ and for Dolma $F = 0.88$ giving us a relative cost of $0.88 + 0.05 = 0.93$.

As can be seen, perplexity pruning leads to a total reduction in cost across all experiments. Furthermore, the cost of reference model training is typically amortized as it's only performed once per dataset. I.e., we used the same reference models to prune the data for both the 1B and 3B models.

## 9 DETAILED EVALUATION SETUP

Jha et al. (2023) also use the MosaicML evaluation gauntlet to perform evaluations in their work. As such, with explicit permission from the authors, we exactly reproduce their text describing the tasks and tasks categories in the evaluation gauntlet. The following is from Section D of their paper:

The **World Knowledge** category includes the following datasets:

- Jeopardy (2,117 questions that are a custom subset of the dataset originally obtained from Wolfe et al. (2022))
- MMLU (14,042 four-choice multiple choice questions distributed across 57 categories Hendrycks et al. (2021)
- BIG-bench wikidata (20,321 questions regarding factual information pulled from wikipedia) Srivastava et al. (2023)
- ARC easy (2,376 easy multiple choice middle school science questions) Clark et al. (2018)
- ARC challenge (1,172 hard multiple choice science questions) Clark et al. (2018)
- BIG-bench: misconceptions (219 true or false questions regarding common misconceptions) Srivastava et al. (2023)

The **Commonsense Reasoning** category loosely assesses a model's ability to do basic reasoning tasks that require commonsense knowledge of objects, their properties, and their behavior. It includes the following datasets:

- BIG-bench Strategy QA (2,289 very eclectic yes/no questions on a wide range of commonsense subjects e.g "Can fish get Tonsilitis?")Srivastava et al. (2023)
- BIG-bench Strange Stories (174 short stories followed by questions about the characters)Srivastava et al. (2023)
- BIG-bench Novel Concepts (32 find-the-common-concept problems)Srivastava et al. (2023)
- COPA (100 cause/effect multiple choice questions) Roemmele et al. (2011)
- PIQA (1,838 commonsense physical intuition 2-choice questions) Bisk et al. (2020)
- OpenBook QA (500 questions that rely on basic physical and scientific intuition about common objects and entities) Mihaylov et al. (2018).

**Language Understanding** tasks evaluate the model's ability to understand the structure and properties of languages, and include the following datasets:

- LAMBADA (6,153 passages take from books - we use the formatting adopted by OpenAI's version)Paperno et al. (2016)
- HellaSwag (10,042 multiple choice scenarios in which the model is prompted with a scenario and choose the most likely conclusion to the scenario from four possible options)Zellers et al. (2019)
- Winograd Schema Challenge (273 scenarios in which the model must use semantics to correctly resolve the anaphora in a sentence. The Eval Gauntlet uses the partial evaluation technique technique introduced in Trinh & Le (2019)) Levesque et al. (2012)
- Winogrande (1,267 scenarios in which two possible beginnings of a sentence are presented along with a single ending) Sakaguchi et al. (2020)
- BIG-bench language identification (10,000 questions on multiple choice language identification) Srivastava et al. (2023)
- BIG-bench conceptual combinations (103 questions using made up words) Srivastava et al. (2023)
- BIG-bench conlang translation (164 example problems in which the model is given translations of simple sentences between English and some fake constructed language) Srivastava et al. (2023)

**Symbolic problem solving** tasks test the model's ability to solve a diverse range of symbolic tasks including arithmetic, logical reasoning, algorithms, and algebra. These datasets include:

- BIG-bench elementary math QA (38,160 four-choice multiple choice arithmetic word problems) Srivastava et al. (2023)
- BIG-bench dyck languages (1000 complete-the-sequence questions) Srivastava et al. (2023)
- BIG-bench algorithms (1,320 questions) Srivastava et al. (2023)
- BIG-bench logical deduction (1500 four-choice multiple choice questions relating to relative ordering of objects) Srivastava et al. (2023)
- BIG-bench operators (210 questions involving mathematical operators) Srivastava et al. (2023)
- BIG-bench repeat copy logic (32 samples in which the model is required to follow some instructions for copying words/symbols)
- Simple arithmetic with spaces (1000 arithmetic problems consisting of up to 3 operations and using numbers of up to 3 digits, developed by MosaicML)
- Simple arithmetic without spaces (1000 arithmetic problems consisting of up to 3 operations and using numbers of up to 3 digits, developed by MosaicML)
- Math QA (2,983 four-choice multiple choice math word problems) Amini et al. (2019)
- LogiQA (651 four-logical word problems) Liu et al. (2020)

The **Reading comprehension** benchmarks test a model's ability to answer questions based on the information in a passage of text. The datasets include:

- BIG-bench Understanding fables (189 short stories) Srivastava et al. (2023)
- Pubmed QA Labeled (1000 hand-labeled medical documents followed by a related question for which the model must respond yes/no/maybe) Jin et al. (2019)
- SQuAD (10,570 short documents followed by a related question. The model is expected to output the exact correct answer) Rajpurkar et al. (2016)
- BoolQ (3,270 short passages on a diverse range of subjects followed by a yes/no questions) Clark et al. (2019)

## 9.1 EVALUATION PROCEDURE

To compute model performance on the above datasets, the Eval Gauntlet uses one of the following three ICL metrics for each dataset (from MosaicML's composer library).

1. InContextLearningQAAccuracy: This metric uses the query, the corresponding correct answer and a list of alternative answers to measure a model's prediction. If the model's response conditioned on the query starts with either the correct answer or with one of the alternative answers, it is considered correct. This is used for question-answering tasks such as TriviaQA.

2. InContextLearningLMAccuracy: This metric tests a model's ability to output a precise set of tokens. A model's output conditioned on a given query is judged to be correct only if the model's highest probability tokens match the correct sequence of tokens. This is used for language modeling tasks such as LAMBADA.

3. InContextLearningMultipleChoiceAccuracy: This metric is used for testing a model's ability to answer multiple choice questions accurately. It compares the respective perplexity of the query prepended to each of the possible choices, according to the model. If the query-choice pair with the lowest per token perplexity is indeed the correct choice, then the model's output is judged to be correct. This is used for multiple choice tasks such as HellaSwag, Winograd etc.

