# OpenReview forum: "Perplexed by Perplexity: Perplexity-Based Data Pruning With Small Reference Models"
_ICLR.cc/2025/Conference — ICLR 2025 Poster_

### Official Review · Reviewer_13P9 · 2024-10-30

**Soundness:** 3
**Presentation:** 3
**Contribution:** 3
**Rating:** 6
**Confidence:** 5

**Summary:**

This paper investigates whether a small model can be used to perform perplexity based data selection for a larger model. The key findings are that 1) a reference model with 30x fewer parameters compared to the larger model can be used to identify a subset of the training data which can improve the performance of the larger model relative to no pruning. 2) the filtered data subset can speed up training of the larger model, 2) the improvements carry over to some extent to over training and data constrained regimes, 3) ideal pruning criteria can vary by dataset e.g. for Pile, a high perplexity subset performs better while for Dolma, a medium perplexity subset works the best. The paper shows that test data perplexity is not a good indicator of the downstream task performance when using perplexity based pruning.

**Strengths:**

* Describes a simple approach to improve the performance of large language models using perplexity based data filtration using a smaller reference model.
* Presents useful results e.g. 1) filtration criteria varies by dataset type and 2) test set perplexity is not a good indicator of the downstream task performance.

**Weaknesses:**

* The main results (Table 1) do not include a random baseline i.e. what is the performance of a model trained on a subset of the data which has a similar size as the perplexity filtered buckets but is selected randomly?
* The paper does not contain ablations on the size of the reference model and sensitivity of the results to the random split (L113) used for training the reference model. Though exploring this space is computationally expensive, it may be useful to present 1-2 additional data points.
* It would be good to see some additional analysis to understand why a high perplexity set works better for one domain while a medium perplexity set works better for others.

Note: The authors have addressed some of these concerns (random baseline/sensitivity to random split) in  the rebuttal.

**Questions:**

* L290: "These results show that while the higher quality data resulting from perplexity-based data pruning does still lead to an improvement in downstream performance in the over-trained regime, there is not a relative increase in downstream improvement over the baseline when over-training." It would be good to understand why this is the case since there are no repeats.
* L314: "That training on repeated perplexity-pruned data leads to diminishing gains after four repetitions post- pruning suggests that the higher quality data resulting from pruning does not change the point for which repeating data yields diminishing improvements in performance." This sentence is confusing and should be reworded.
* In section 4.2, the paper presents results showing that the pruning affects data composition such that some domains (e.g. web) are oversampled compared to others (e.g. pubmed). It would be useful to perform additional analysis to understand why this is the case e.g. is it possible that the training split (L113) resulted in a smaller proportion of these domains for the reference dataset?

---

> ### Author Response · Authors · 2024-11-22
> **Response to Reviewer 13P9 (part 1/2)**
>
> We would like to thank Reviewer 13P9 for the time they spent reviewing our paper. Our responses to the reviewer’s feedback are listed below.
>
> > “The main results (Table 1) do not include a random baseline i.e. what is the performance of a model trained on a subset of the data which has a similar size as the perplexity filtered buckets but is selected randomly?”
>
> The baseline results presented in Table 1 are implemented as described in the reviewer’s comment as our pruning results are constructed such that the number of tokens post-pruning equals the training duration. Namely, if our experiment has a selection rate of $r_s$ and we have a desired training duration of $D$ examples, then we compute the perplexity and sample from a pool of  $\frac{1}{r_s}D$ examples from the original dataset. As such, post-pruning we are left with $D$ examples. The baseline results in Table 1 are constructed by just training on a random set of $D$ examples from the original dataset. As such, the baseline is a random selection that is the same size as the perplexity-pruned data. We understand that this is not stated clearly in the paper, and we will clarify these details in our experimental setup.
>
> The results in Table 1 are conducted under the assumption that the dataset is sufficiently large such that we can prune without requiring multiple passes through the pruned data. We relax this assumption in Section 3.5 where we investigate the data-constrained regime, and the baseline results in Figure 2 are randomly sub-sampled such that they are of the same size as the perplexity pruned data.
>
> > “The paper does not contain ablations on the size of the reference model and sensitivity of the results to the random split (L113) used for training the reference model. Though exploring this space is computationally expensive, it may be useful to present 1-2 additional data points.”
>
> We agree that evaluating the impact of reference model size on final performance is an interesting direction for future work. Based on the findings of Marion et al. [1] we expect that increasing the reference model size would lead to increases in pruning performance, but it would indeed be valuable to reconfirm this for our experimental setup. We will add this result for the final paper.
>
>  The results we present in the paper do factor in the sensitivity to the random split used to train the reference model. Each experiment in the paper is conducted across two seeds, and the same random seed used for training the final model is also used to split the dataset and train the reference model. Thus the gain’s from perplexity pruning are robust to the random split used as each result presented in the paper is the performance across two different random splits of the training data. We will clarify this detail in our experimental setup for the final paper.
>
> > “It would be good to see some additional analysis to understand why a high perplexity set works better for one domain while a medium perplexity set works better for others.”
>
> Developing a better understanding for why different domain compositions admit different optimal pruning strategies is an interesting direction for future work. One possible explanation can be seen from the fact that the post-pruning domain compositions become more similar (Figure 4). Namely, for both the Pile and Dolma the web crawled domains (cc, openwebtext2, etc.) are upweighted while specialized domains (pubmed-central, the stack) are downweighted suggesting that in both cases it is optimal to have a larger portion of web-data. In order to prune to more web data on the Pile, high perplexity samples must be selected as there is a large proportion of specialized domains which typically have samples with lower perplexity. For Dolma on the otherhand, it is primarily composed of web-data, and as such the medium perplexity samples are retained. We will make this discussion more explicit in the final version of the paper.

---

> > ### Author Response · Authors · 2024-11-22
> > **Response to Reviewer 13P9 (part 2/2)**
> >
> > > “L290: "These results show that while the higher quality data resulting from perplexity-based data pruning does still lead to an improvement in downstream performance in the over-trained regime, there is not a relative increase in downstream improvement over the baseline when over-training." It would be good to understand why this is the case since there are no repeats.”
> >
> > While we agree that such an analysis would be interesting, we believe that it is out of scope for our current work. We would also like to emphasize that the comment in the paper is about the relative delta between pruned performance and baseline performance not improving. Importantly, the absolute gain over the baseline is still largely preserved even as we increase the data budget. That the absolute gain from pruning is fixed regardless of training duration is also consistent with the downstream evaluations conducted at different checkpoints during training (Figure 1). With those results, we find that there is a mostly constant gain from training on pruned data throughout the whole duration of training.
> >
> > > “L314: "That training on repeated perplexity-pruned data leads to diminishing gains after four repetitions post-pruning suggests that the higher quality data resulting from pruning does not change the point for which repeating data yields diminishing improvements in performance." This sentence is confusing and should be reworded.”
> >
> > Thank you for bringing this to our attention. In the final version of the paper we will change the wording to be:
> >
> >
> > “Although one might hope that training on higher quality pruned data would allow for more repetitions through the data without saturation, we empirically find that this is not the case.”
> >
> > > “In section 4.2, the paper presents results showing that the pruning affects data composition such that some domains (e.g. web) are oversampled compared to others (e.g. PubMed). It would be useful to perform additional analysis to understand why this is the case e.g. is it possible that the training split (L113) resulted in a smaller proportion of these domains for the reference dataset?”
> >
> > We investigated whether the domain compositions of the random splits were skewed and we found that the proportion of all random splits was within +- 1% of the original proportions. As the upweighting of web domains is observed across both the Pile and Dolma for two different pruning strategies, we are inclined to believe that the increased performance of web-domains is a more general phenomenon. While we agree that developing a deeper understanding of why web domain data leads to better LLM performance, we believe it is outside of the scope of our research.
> >
> >
> > ## References
> > [1] Marion, Max, et al. "When less is more: Investigating data pruning for pretraining llms at scale." arXiv preprint arXiv:2309.04564 (2023).

---

> > > ### Comment · Reviewer_13P9 · 2024-11-22
> > >
> > > I appreciate the clarifications and thank the authors for taking the time to write the response.

---

> > > > ### Author Response · Authors · 2024-11-25
> > > >
> > > > We greatly appreciate reviewer 13P9 for engaging with the rebuttal and reading our responses. We are pleased to know that the clarifications were helpful for the reviewer. We would just like to follow up and see if there are any other questions we can answer or information we can provide that would convince you to increase our scores/advocate for acceptance of the paper.

---

### Official Review · Reviewer_wYdV · 2024-11-01

**Soundness:** 3
**Presentation:** 3
**Contribution:** 3
**Rating:** 6
**Confidence:** 3

**Summary:**

The authors filter LLM pre-training data by using the perplexity of a smaller language model. They demonstrate that dataset filtering improves the [initial] learning curve of LLM pre-training.

**Strengths:**

The method is well motivated. Except for some uncommon terminology that is explained in later sections like "non-standard training regime", "over-training" (which is not over-fitting) the paper is clearly written.

**Weaknesses:**

L186 suggests that the final models are (pre-)trained for a fixed number of steps, no matter the dataset size. This sets the stage for dataset filtering, since training on the full dataset may go through fewer epochs. It would be interesting to train for long enough to show convergence in the plots in Fig. 1.  The story would be more convincing if there is an offset between the blue and red curves even after convergence. In fact, the "over-training" experiment in Sec. 3.4 shows diminishing gains, so I can imagine that they disappear fully at some point. The method would still have merits (steeper pre-training curve), just not the ones claimed in the paper.

Novelty. Perplexity-based pruning and countless variations of it are well-studied. The authors set their work apart from prior work in L058, but neither of the arguments (i)-(iii) (evaluation on downstream task, exploration of domain compositions, "non-standard" evaluation regimes) strike me as particularly strong.

I don't think that Algorithm 1 is really helping clarity. 1-2 normal equations would be just as expressive and more concise.

Edit: my point about novelty was unjustified - I have increased my scores after the rebuttal

**Questions:**

- Fig.4 is interesting, but I'm not sure how Fig. 3 is relevant in practice - could you clarify?

---

> ### Author Response · Authors · 2024-11-22
> **Response to Reviewer wYdV**
>
> We would like to thank Reviewer wYdV for their review and the questions that they asked. We also appreciate that the reviewer finds the method to be “well motivated”. Our responses to the reviewer’s feedback are listed below.
>
> > “It would be interesting to train for long enough to show convergence in the plots in Fig. 1. The story would be more convincing if there is an offset between the blue and red curves even after convergence. In fact, the "over-training" experiment in Sec. 3.4 shows diminishing gains, so I can imagine that they disappear fully at some point. The method would still have merits (steeper pre-training curve), just not the ones claimed in the paper.”
>
> While training to convergence would be an interesting data point, we don’t believe this is computational feasible with any reasonable budget. Consider the experiments in which we trained for 5x the Chinchilla training duration for the 1B models. We have plotted the performance at intermediate checkpoints throughout training as done in Figure 1 for the 1B parameter model trained 5x chinchilla on the Pile, and the resulting plot can be accessed [here](https://postimg.cc/vgfBGsBZ). As can be seen, even training for 5x the Chinchilla training duration does not saturate the performance.
>
> We would like to make the further meta point that training to convergence is not the standard practice when training LLMs and the standard is training a model for the compute optimal duration [1]. Additionally, in the age of scaling when one benchmark saturates we evaluate on harder benchmarks instead, and as such the performance people care about is not in the convergened regime.
>
> > “Novelty. Perplexity-based pruning and countless variations of it are well-studied. The authors set their work apart from prior work in L058, but neither of the arguments (i)-(iii) (evaluation on downstream task, exploration of domain compositions, "non-standard" evaluation regimes) strike me as particularly strong.”
>
>
> We are only aware of one paper that examines model-based perplexity pruning for pretraining LLMs before ours [2] and as such we believe that it is a false characterization to say this setting is “well-studied”. With regard to this earlier paper, the differences we outline are very significant. Without evaluating based on downstream evaluations, one would conclude that the technique does not work unless the reference model is significantly larger than the final model. This conclusion would severely limit the applicability of perplexity-based data pruning as it would only be a useful technique for training smaller language models.
>
> We also strongly believe that evaluating multiple domain compositions and non-standard training regimes have significant implications. By evaluating multiple domain compositions, our research is actually applicable for practitioners as they can choose the proper pruning setting based on their dataset composition. As stated, we find that the optimal settings for one domain composition may actually lead to worse performance than no pruning on another composition. Furthermore, by evaluating the over-trained and data-constrained settings, we provide the first guidance on when the technique should be expected to work in non-standard settings.
>
> > “I don't think that Algorithm 1 is really helping clarity. 1-2 normal equations would be just as expressive and more concise.”
>
> Thank you for this feedback. While we do believe that there are some important details conveyed in the algorithm that would be harder to communicate in text, we agree that it is more complex than necessary. We will simplify the algorithm in the final paper.
>
> > “Fig.4 is interesting, but I'm not sure how Fig. 3 is relevant in practice - could you clarify?”
>
> The purpose of Figure 3 is to provide readers with intuition for both the differences in the distribution of text between the Pile and Dolma and to demonstrate what effect pruning has on the distributions. Namely, the Pile is composed of many domains and as such its distribution has multiple modes while Dolma is predominantly a single domain and correspondingly unimodal. Additionally, we make the point that the perplexity distribution of both datasets has a very similar shape post-pruning. This suggests a potential reason why different pruning strategies are superior on different domain compositions, as different pruning strategies are needed to achieve similar perplexity distributions post-pruning.
>
> ## References
> [1] Hoffmann, Jordan, et al. "Training compute-optimal large language models." arXiv preprint arXiv:2203.15556 (2022).
>
> [2] ​​Marion, Max, et al. "When less is more: Investigating data pruning for pretraining llms at scale." arXiv preprint arXiv:2309.04564 (2023).

---

### Official Review · Reviewer_wjeV · 2024-11-03

**Soundness:** 3
**Presentation:** 4
**Contribution:** 3
**Rating:** 6
**Confidence:** 4

**Summary:**

This paper presents a perplexity-based pruning method for reducing the size of pre-training datasets. The effect of pruning is evaluated through the performance on downstream tasks as well. Two datasets are used for evaluation: Pile and Dogma. The pruning efficacy is determined for over-trained and data-constrained regimes as well.

**Strengths:**

- The paper addresses an important problem of pruning the pre-training datasets to enable efficient training of LLMs.
- The experiments are thorough and cover different dimensions of perplexity-based pruning.
- The paper is well-written and the results are presented clearly.
-  The findings are significant, as they show that perplexity-based data filtering can not only reduce the size of the pre-training datasets, it also leads to better performance on certain downstream tasks.

**Weaknesses:**

- The paper does not currently cover the computational complexity of the proposed pruning procedure. A few important questions that need to be considered in this regard:
    - How do the computational requirements for perplexity-based pruning increase with the size of the dataset to be pruned?
    - How does the cost of computing perplexity (before pruning) amortize over the efficiency improvements achieved while pretraining the model on the pruned datasets?
- A discussion for choosing the right perplexity pruning method (low, medium, high) for the dataset should be included for the practitioners. From the experimental results, we can see that high perplexity selection performs better on Pile while medium perplexity selection is better for dolma. Can we extract any patterns from these results and other experiments that can be generalized to other datasets?
    - For example, prior theory on data pruning for vision tasks shows that the optimal pruning strategy changes depending on the amount of initial data. When data is abundant, the better pruning strategy is to keep harder examples. In contrast, for smaller datasets, keeping the easier examples leads to better performance. [1]
- The results show that test set perplexity may not always be a sound metric for evaluating a pruning strategy and that downstream evaluation is necessary. What should be the cheapest way of conducting the downstream evaluation of the correct perplexity pruning method, i.e., the one that can yield reliable results at a minimal cost? For example, could there be a small set of representative downstream tasks or metrics that could serve as efficient proxies for full downstream evaluation?

References:

[1] https://arxiv.org/abs/2206.14486

**Questions:**

- A quantized model may lead to better inference efficiency while calculating the perplexity. Was this considered while running the experiments?
- High perplexity selection will also inevitably lead to the inclusion of a significant portion of the noisier examples in the overall dataset. How can we determine the proportion of such examples in the final dataset and exclude them reliably?
- Minor typo (line 66): perplexity-basd -> perplexity-based
- It would be useful to include the following closely related data pruning works in the related work section:
    - https://arxiv.org/abs/2403.07384
    - https://arxiv.org/abs/2402.09668

---

> ### Author Response · Authors · 2024-11-22
> **Response to Reviewer wjeV (part 1/2)**
>
> We would like to thank Reviewer wjeV for the time that they spent reviewing our paper. Our responses to the reviewer’s feedback are listed below.
>
> > “The paper does not currently cover the computational complexity of the proposed pruning procedure. A few important questions that need to be considered in this regard:
> > * How do the computational requirements for perplexity-based pruning increase with the size of the dataset to be pruned?
> > * How does the cost of computing perplexity (before pruning) amortize over the efficiency improvements achieved while pretraining the model on the pruned datasets?”
>
> We will add a discussion of the compute requirements to the paper as we agree it is important. Assuming that the reference model is fixed (i.e., the same size reference model trained on the same number of tokens) as we do, then the only compute required is for performing inference from the reference model over the dataset to be pruned. So the pruning compute grows as $O(N_{ref} \frac{D}{r_s})$ where $N_{ref}$ is the number of reference parameters, $r_s$ is the selection rate, and $D$ is the number of training tokens. As $N_{ref} << N_{final}$, this operation is relatively cheap compared to the overall cost.
>
> We now turn towards the total computational complexity of perplexity pruning. As our method does not change the computation performed on the forward or backward pass, and only affects which tokens are trained on, we can analyze the compute requirements in terms of the total number of operations. We will approximate the cost of training as $6ND$ where $N$ is the number of parameters and $D$ is the number of training tokens, and we will approximate the cost of computing a sequence’s perplexity in inference mode as $2ND$ [1][2]. Assuming a reference model of size $N_{ref}$, number of reference training tokens $D_{ref}$, final model size $N_{final}$, number of final tokens $D_{final}$, selection rate $r_s$, and fraction of tokens for the pruned data to achieve the same performance as the baseline $F$, the relative cost of perplexity pruning compared to the baseline is:
>
> $$
> \frac{2N_{ref} \frac{D_{final}}{r_s} + 6N_{ref} D_{ref} + 6N_{final} FD_{final}}{6N_{final}D_{final}} = \frac{N_{ref}}{N_{final}}(\frac{1}{3r_s} + \frac{D_{ref}}{D_{final}}) + F
> $$
>
> All our reference models are 125M parameters, trained for 26B tokens, and we use $r_s = 0.5$ throughout. The 1B models are trained on 26B tokens giving us $\frac{N_{ref}}{N_{final}}(\frac{1}{3r_s} + \frac{D_{ref}}{D_final}) = \frac{125 \times 10^6}{1.3 \times 10^9}(\frac{2}{3} + \frac{26 \times 10^9}{26 \times 10^9}) = 0.16$.
> For the Pile, $F=0.76$ giving us a relative cost of $0.16 + 0.76 = 0.92$ and for Dolma $F=0.78$ giving us a relative cost of $0.16 + 0.78 = 0.94$.
> The 3B models are trained on 54B tokens giving us $\frac{N_{ref}}{N_{final}}(\frac{1}{3r_s} + \frac{D_{ref}}{D_final}) = \frac{125 \times 10^6}{2.7 \times 10^9}(\frac{2}{3} + \frac{26 \times 10^9}{54 \times 10^9}) = 0.05$.
> For the Pile $F=0.69$ giving us a relative cost of $0.69 + 0.05 = 0.74$ and for Dolma $F=0.88$ giving us a relative cost of $0.88 + 0.05 = 0.93$.
> As can be seen, perplexity pruning leads to a total reduction in cost all experiments. We would also like to emphasize that the cost of reference model training is typically amortized as it's only performed once per dataset. I.e. we used the same reference models to prune the data for both the 1B and 3B models.
>
> > “A discussion for choosing the right perplexity pruning method (low, medium, high) for the dataset should be included for the practitioners. From the experimental results, we can see that high perplexity selection performs better on Pile while medium perplexity selection is better for Dolma. Can we extract any patterns from these results and other experiments that can be generalized to other datasets?”
>
> We agree that such a discussion would be useful and will add it to the final paper. Our results are still useful for practitioners without needing to extrapolate any trends as the Pile and Dolma cover the two primary types of domain compositions for pretrainnig datasets. Namely, datasets are either composed of lots of specialized, skilled domains or predominantly general web scrapes. As we test both these settings, we believe practitioners can use the same settings we find depending on which of the two types of dataset compositions they have.

---

> ### Author Response · Authors · 2024-11-22
> **Response to Reviewer wjeV (part 2/2)**
>
> > “A quantized model may lead to better inference efficiency while calculating the perplexity. Was this considered while running the experiments?”
>
> We did not consider such a method for reducing the perplexity calculation costs at the time of writing the paper but we suspect that it would work and should be used in future experiments.
>
> > “High perplexity selection will also inevitably lead to the inclusion of a significant portion of the noisier examples in the overall dataset. How can we determine the proportion of such examples in the final dataset and exclude them reliably?”
>
>
>
> We agree that while selecting higher perplexity samples leads to an improvement in performance, it may bias the dataset to contain noisier samples. One potential method for removing such examples is combining our perplexity-based pruning method with other data filtering methods specifically targeted at noisy examples. One example would be training a simple classifier on curated examples of noisy text [4] and then using that classifier to further prune the perplexity-pruned dataset.
>
>
>
>
> > “Minor typo (line 66): perplexity-basd -> perplexity-based”
>
> Thank you for catching this. It is now fixed.
>
>
>
> > “It would be useful to include the following closely related data pruning works in the related work section.”
>
> Thank you for bringing these related works to our attention. We will include the updated related works in the final version of our paper.
>
> ## References
>
> [1] Kaplan, Jared, et al. "Scaling laws for neural language models." arXiv preprint arXiv:2001.08361 (2020).
>
> [2] Anthony, Quentin, et al. "Transformer Math 101." EleutherAI, 18 Apr. 2023, blog.eleuther.ai/transformer-math/.
>
> [3] Barton, Tessa. "Calibrating the Mosaic Evaluation Gauntlet." Databricks, 30 Apr. 2024, https://www.databricks.com/blog/mosaic-research/calibrating-mosaic-evaluation-gauntlet.
>
> [4] Gao, Leo. "An empirical exploration in quality filtering of text data." arXiv preprint arXiv:2109.00698 (2021).

---

> > ### Comment · Reviewer_wjeV · 2024-11-24
> >
> > Thank you to the authors for their detailed response. The results regarding computational complexity has addressed my concerns.

---

> > > ### Author Response · Authors · 2024-11-24
> > >
> > > We greatly appreciate the reviewer for engaging with the rebuttal and reading our responses. We are pleased to know that the discussion regarding computational complexity addressed the reviewer's concerns. We would just like to follow up and see if there are any other questions we can answer or information we can provide that would convince you to increase our scores/advocate for acceptance of the paper.

---

### Official Review · Reviewer_qYWL · 2024-11-05

**Soundness:** 3
**Presentation:** 3
**Contribution:** 3
**Rating:** 8
**Confidence:** 3

**Summary:**

The paper proposes that smaller language models effectively prune large datasets in a way that benefits the training of much larger model. Applying perplexity-based pruning techniques, they explore using a small model to filter high-quality subsets of data for training larger models. This approach is interesting because it’s a cost-effective alternative to using large models for pruning, and is applicable in real settings. The findings indicate benefits for downstream accuracy and training efficiency.

The paper demonstrates that a 125m parameter model can successfully prune data for large models and improve downstream task performance. The paper shows empirical results testing on The Pile and Dolma, two datasets with very different domain structures.
They also study the two settings of over-training and data-constrained setups and provide additional insights.

**Strengths:**

The goal, and the process, and algorithm are defined and presented very clearly. Experiments cover multiple settings, with different model sizes and training algorithms.
The proposed method is super useful for researchers who investigate practical techniques for data curation, with insightful empirical results.
Experiments include two very different dataset distributions, the Pile dataset and Dolma. The work shows thorough experiments for various selection rates and perplexity criteria, presenting strong evidence about settings in which perplexity pruning does and does not work.

**Weaknesses:**

Authors claim that datasets pruning increases the proportion of general domain data from web-scraped domains, and decreases the proportion of specific and technical domains. But it is unclear and counter intuitive why training on general domain data improves performance of models on benchmarks. I think the paper lacks analysis to explain this observation.

**Questions:**

How do you expect the results to scale on models larger than 3B parameters?

How does models' performance change on domains which are pruned the most?

---

> ### Author Response · Authors · 2024-11-22
> **Response to Reviewer qYWL**
>
> We would like to thank Reviewer qYWL for their detailed review of the paper and for finding the proposed method “super useful”. Our responses to the reviewer's questions are listed below.
>
> > “Authors claim that datasets pruning increases the proportion of general domain data from web-scraped domains, and decreases the proportion of specific and technical domains. But it is unclear and counter intuitive why training on general domain data improves performance of models on benchmarks. I think the paper lacks analysis to explain this observation.”
>
> As the upweighting of web domains is observed across both the Pile and Dolma for two different pruning strategies, we are inclined to believe that the increased performance of web-domains is a more general phenomenon of llm pretraining. While we agree that developing a deeper understanding of why web domain data leads to better LLM performance, we believe it is outside of the scope of our research.
>
>
>
> > “How do you expect the results to scale on models larger than 3B parameters?”
>
> While we believe that using a 125M parameter reference model to prune data for the Pile would continue to yield gains past 3B parameters, there is too much uncertainty to make any claims for Dolma. On the Pile, the gap between the average pruned performance and the average baseline performance actually grows from 1.89 to 2.04 for the 1B and 3B parameter final models respectively. However, on Dolma the gap between the average pruned performance and the average baseline performance shrinks from 1.51 to 0.59 for the 1B and 3B parameter final models respectively.
>
> We would like to emphasize that this is not to say that we don’t believe the method would for models larger than 3B on Dolma. Even if using a 125M parameter reference model stopped showing gains past 3B final parameters, we believe that through scaling the size of the reference model proportionality to the final model size we would continue to see gains on Dolma.
>
> > “How does models' performance change on domains which are pruned the most?”
>
> The model sizes and token budgets we evaluate aren’t sufficiently large to get signal on the domains that are primarily pruned (i.e., code tasks). The closest category from the evaluation gauntlet that we use is the “Symbolic Problem Solving” category. As seen in Table 1, while there is no statistically significant difference for the 1B models, the baseline models outperform the models trained on pruned data for the symbolic problem-solving category at the 3B model scale. We hope to fully investigate how pruning affects coding performance at scale in future work. If performance does degrade on coding tasks, we believe that there would be promising strategies to mitigate this such as domain upsampling at the end of training [1] or applying pruning on a per-domain basis.
>
>
>
> ## References
> [1] Blakeney, Cody, et al. "Does your data spark joy? Performance gains from domain upsampling at the end of training." arXiv preprint arXiv:2406.03476 (2024).

---

> ### Comment · Reviewer_qYWL · 2024-11-26
>
> Based on your response, since you do not have the capacity to empirically test this for the paper, I think it is critical to mention the unknown effect of pruning on performance in the pruned domains in your paper.

---

> > ### Author Response · Authors · 2024-11-26
> >
> > We thank the reviewer for reading our responses and for their further suggestion. We agree that such a comment is important to include in the paper and will add the following to the Discussion section of the final paper:
> >
> > > “One limitation of our work is that while domains such as code are largely pruned, we do not have the computational budget required for training models of sufficient size to test the coding ability of models on benchmarks such as HumanEval [1]. It remains important future work to further scale perplexity pruning to larger models such that the impact of pruning specific domains can be better understood.”
> >
> > ## References
> >
> > [1] Chen, Mark, et al. "Evaluating large language models trained on code." arXiv preprint arXiv:2107.03374 (2021).

---

### Meta-Review · Area_Chair_rnXz · 2024-12-20

**Metareview:**

This paper presents a study on using smaller language models to prune large datasets effectively, aiming to improve the training efficiency of larger models. The proposed perplexity-based pruning technique is evaluated on two distinct datasets, The Pile and Dolma, which vary in domain structure. The study encompasses different experimental settings, namely over-training and data-constrained setups. The paper concludes that smaller models can successfully select high-value data subsets, which enhance training efficiency and downstream accuracy for larger models.

This paper tackles a significant challenge in natural language processing: the efficient training of LLMs through dataset pruning. By leveraging smaller models for perplexity-based data pruning, the proposed method is a cost-effective alternative to using large models. The extensive empirical results across varied datasets and experimental settings provide persuasive evidence of the effectiveness and practical applicability of this approach. The paper's insights into data composition effects and pruning strategies enrich the understanding of dataset management in model pre-training, making it a valuable contribution to the field.

Suggestions:
1. A deeper analysis is recommended to clarify why training on general domain data enhances model performance on benchmarks, despite domain-specific prunings which seem counterintuitive. Although the authors note this is outside their current scope, addressing this point would enhance the paper's comprehensiveness.
2. The paper should include a discussion of the computational requirements regarding perplexity pruning complexity. While responses address this, embedding it in the paper would guide practitioners better.
3. A clearer presentation of the random baseline setup is vital, as well as an explicit clarification in the experimental setup, to make these comparisons transparent. Some further ablation studies or sensitivity analysis related to the reference model size and pruned datasets would benefit the study, albeit acknowledged as resource-intensive.
4. Simplifying some parts of the paper, such as Algorithm 1, will assist in enhancing the accessibility of the methodologies delineated.

**Additional Comments On Reviewer Discussion:**

The reviewers initiated several insightful discussions, specifically about the novelty of using perplexity-based pruning for LLMs, differences in pruned dataset compositions, and strategies based on dataset domains. Reviewers raised questions about computational complexities, methodological clarifications, and downstream continuum for over-trained scenarios. Through author's responses, many areas were clarified, which significantly improved reviewer understandings. The discourse notably resulted in an improved appreciation and alignment of reviewer scores on the practical novelties presented by the study, specifically after rebuttal clarifications around benchmarking without convergence due to resource constraints.

---

### Decision · Program_Chairs · 2025-01-22

Accept (Poster)